# Psychometric Properties and Effects on Health Outcomes of the Patient Assessment of Chronic Illness Care (PACIC) in Korean Hemodialysis Patients

**DOI:** 10.3390/healthcare10061149

**Published:** 2022-06-20

**Authors:** Ae-Rim Seo, Bo-Kyoung Kim, Ki-Soo Park

**Affiliations:** Department of Preventive Medicine and Institute of Health Science, College of Medicine, Gyeongsang National University, Jinju 52727, Korea; sarim2101@naver.com (A.-R.S.); tomato9804@gmail.com (B.-K.K.)

**Keywords:** health outcomes, chronic care model, patient assessment, hemodialysis

## Abstract

Background: The satisfaction of patients receiving integrated care with End-Stage Renal Disease (ESRD) is widely advocated and patients with ESRD have special health needs, but few studies have investigated whether integrated care was associated with health outcomes. Our aims were to evaluate the psychometric properties of the Korean-translated patient assessment of chronic illness care (PACIC) in patients with ESRD, and to evaluate whether PACIC evaluated by patients was associated with health outcomes. Methods: ESRD patients on hemodialysis (n = 172) at 2 dialysis centers. Data quality, internal consistency and correlation between items and scales were assessed. To test the external validity, the association between PACIC and the health behaviour and outcomes of hemodialysis patients was analyzed. Results: The mean score of the PACIC items was 3.0. The item-scale correlation (0.67–0.85) and test-retest correlation (0.72–0.82) regarding scales for internal consistency showed excellent consistency. Total PACIC score was significantly associated with dietary self-efficacy (β = 0.22) and serum potassium (Exp(B) = 1.65). Higher overall PACIC score was significantly associated with higher physical health status (β = 3.52). Conclusions: The Korean-translated PACIC questionnaire is a tool with reliability and validity. Comprehensive treatment strategies for ESRD patients may improve their health behaviors and outcomes.

## 1. Introduction

The incidence of chronic kidney disease (CKD) continues to increase worldwide despite progress in integrated care [1,2]. CKD has become an increasing health problem globally, both in terms of mortality and healthcare costs. In order to improve patient outcomes, more integrated treatment is needed for end-stage renal disease (ESRD) [2].

Patients with ESRD also demand that the healthcare system is changed to provide comprehensive, proactive, self-care management support, also known as the chronic care model (CCM). CCM was developed to improve the quality of care for chronically ill patients. The CCM contains the core contents of self-management support and planned, proactive, population-based care [3,4].

The Assessment of Chronic Illness Care (ACIC) [5] and the Patient Assessment of Chronic Illness Care (PACIC) [6] tools were developed to assess CCM implementation at the provider and patient levels, respectively. The PACIC consists of 20 items as a tool to evaluate the quality of care experienced by patients over the past 6 months [6]. The English version of the PACIC was validated for a variety of chronic diseases (e.g., patients with diabetes, hypertension and other ailments) and population groups [7,8,9]. In diabetic patients [10,11], PACIC had been shown to be associated with increased physical activity and appropriate laboratory assessment and self-management. However, there are no studies on the relationship between PACIC and self-management and health outcomes in hemodialysis patients, and it has not yet been attempted to translate it into Korean.

This study aimed to guide the assessment of associations between health outcomes using Korean-translated PACIC to evaluate a patient-centered communication model in ESRD patients.

## 2. Materials and Methods

### 2.1. Translation of the PACIC Questionnaire

The Korean version of the PACIC was translated and retranslated according to guidelines for cultural adaptation. First, the original English version of the PACIC was translated into Korean by a professional translator who is a native Korean speaker that majored in English literature. Our panel consisted of two nephrologists and two epidemiologists who reviewed the preliminary Korean version of the PACIC and revised it to be as similar as possible in terms of wording and meaning to the original English version. Two additional bilingual doctors back-translated this revised Korean version of the PACIC. The correspondence between this back-translated version and the original English version was reviewed by the research team.

Next, the Korean-translated PACIC was tested in 5 patients on hemodialysis. This was primarily a procedure for simplifying and clarifying the questions asked. We assessed whether the patients understood all items, even items that the interviewer thought were clear to the patients. Some questions required cultural adaptation, such as item 18 ‘referred to a dietician, health educator or counsellor’. In the physician-centered Korean healthcare system, dietary services and health education were usually provided by doctor or by a specially-trained nurse or dietician.

As in the original version, the patients responded to 20 items related to how often they received care according to the integrated care. Each item was on a 5-point Likert scale (1 (no/never) to 5 (full accordance; yes/always)). The total average score was 1–5, with higher scores indicating more relevant areas of comprehensive chronic disease management. This tool had five domains: patient activation (3 items), delivery system/practice design (3 items), goal setting/tailoring (5 items), problem solving/contextual (4 items) and follow-up/coordination (5 items).

### 2.2. Study Population

Among ESRD patients (189 patients) receiving hemodialysis at 2 dialysis centers in Korea, 172 patients who did not meet the exclusion criteria and agreed to the study were included in the study. Exclusion criteria were dialysis patients due to acute kidney injury and had severe stroke. For patients with reading and writing difficulties, the questionnaire was read out loud and filled out during an oral interview. To evaluate test-retest reliability, the second survey was handed out 1 week later to 45 patients who were selected randomly using SPSS v. 25.0. The number of test-retest subjects was arbitrarily set to 45 in order to be conducted within 20–30% of the study population. Also, an explanatory note was attached that the second questionnaire that was not intended to test whether the patient remembered the initial survey. This 1-week follow-up period was much shorter than the interval proposed by Glasgow et al. [6]. The reason that the retest period was shortened is that the longer the interval, the more changes in lifestyle or treatment, and these changes may bias the reliability. Forty-three out of 45 individuals responded to the retest. The Institutional Review Board of Kyungpook National University Hospital approved the research protocol. All subjects provided written informed consent.

### 2.3. Measures

We analyzed the following socio-demographic and clinical characteristics of the patients: sex; age; educational level; job; dialysis duration; cause of ESRD; co-morbidity; laboratory data. As an estimation of external validity, we used dietary self-efficacy, clinical data and health related quality of life.

Dietary self-efficacy was validated and included 14 items [12]. We only used the situational self-efficacy of compliance with dietary guidelines (10 items) because this was significantly associated with serum potassium. All items were scored on a 5-point Likert-type scale with scores that ranged from strongly agree (1) to strongly disagree (5). Cronbach’s alpha, a measure of the internal consistency of the scale, was 0.93 in this study population. The mean ± SD dietary self-efficacy score was 3.5 ± 0.72.

The main clinical data, such as pre-dialysis serum potassium, phosphorus, albumin and interdialytic weight gain, as well as medical history, were also assessed. Clinical data was examined using a dichotomous variable (serum potassium level < 5.0 or ≥ 5.0 mEq/L; serum phosphorus level < 4.5 or ≥ 4.5 mEq/L; serum albumin level < 4.0 or ≥ 4.0 g/dL; interdialytic weight gain < 2.0 or ≥ 2.0 kg).

As a health-related quality of life, KDQOL-36 was used. The Korean version of the KDQOL-36 includes 12 items that provide a generic core (SF-12), as well as 24 additional items. The 24 additional items were related to particular health-related concerns of individuals with kidney disease (i.e., symptom/problem list, 12 items; effects of kidney disease, 8 items; and burden of disease, 4 items). The KDQOL-36 scores had a 0–100 possible range, and the higher scores, the better the health-related quality of life.

### 2.4. Analyses

The psychometric elements of the PACIC were examined in two parts. First, we assessed the data quality (mean, ceiling and flooring effect), internal consistency and correlation between items and scales and intra-class coefficient (ICC) for test-retest reliability. Second, to verify the external validity, we used multiple linear (or logistic) regression analysis with dialysis staff encouragement scores, and to identify the associations between dietary self-efficacy, clinical data and HRQOL as dependent factors and enablement or evaluation as an independent factor (PACIC), we also used multiple linear (or logistic) regression analysis. Age, sex, educational level, job, dialysis duration, cause of ESRD and co-morbidity were also included in the model for adjustment.

## 3. Results

Table 1 shows the demographics and laboratory values of the study population. Of the 172 patients, 81 (47.1%) were women, the mean age was 59.2 ± 12.9 years and just 28.5% were employed. The overall PACIC value of the study subjects was 3.0 ± 0.8. None of the demographic factors, such as sex, age, cause of ESRD or comorbidities, were associated with the overall value. Eighty-three patients (48.3%) showed pre-dialysis hyperkalaemia. The overall value was significantly higher in patients with serum potassium levels in the normal range (3.1 ± 0.7) than in patients with hyperkalaemia (2.8 ± 0.8). However, there were no significant differences in the overall PACIC score between patients with other clinical measures in the normal range versus values that were out of the normal range.

Table 2 shows the descriptive statistics of the individual scales: mean, standard deviation, floor effects and ceiling effects. The total mean score of the original PACIC items was 3.0. There were no notable floor effects or ceiling effects; both were below 6.4%. Regarding scale for internal consistency, the correlations of single items and the referring scale (item–scale correlation) ranged from 0.67 to 0.85. Cronbach’s alpha was 0.83 for the overall PACIC. Values for the scales were at least 0.65 (patient activation scale). The test-retest reliability values were high, achieving the highest value for the follow-up/coordination scale (0.81) and the lowest for the goal setting/tailoring problem scale (0.72).

The results of multiple regression analysis between the PACIC scales and dietary efficacy scores, potassium levels and HRQOL are displayed in Table 3. The independent associations between dietary self-efficacy and PACIC score was significant (β = 0.22, *p* < 0.001). The independent associations between the serum potassium level, HRQOL and PACIC were all significant (OR = 1.65, *p* = 0.040, β= 3.52, *p* = 0.049). However, the overall PACIC scores were not significantly associated with serum phosphorus, albumin and interdialytic weight gain and kidney-disease-targeted items (symptom/problem, effects of kidney disease, and burden of disease) in multiple analysis (data not shown).

## 4. Discussion

In this study, we translated and validated the original 20-item PACIC scale into a Korean version. The PACIC should not be related to socio-demographic variables, as noted by the developers [6]. Indeed, we found that the PACIC scores in Korean hemodialysis patients were not associated with socio-demographic characteristics such as sex, age, level of education and job. This suggests that the translation was successful despite possible variations in other unmeasured factors, such as cultural aspects and health literacy.

The Korean-translated PACIC questionnaire demonstrated high reliability, internal consistency and test-retest reliability. This instrument is expected to be applicable to Korean hemodialysis patients. Its items proved to be selective and non-redundant, as reflected by very satisfactory results for Cronbach’s alpha. The values for ICC, which assesses test-retest reliability, indicated good reproducibility. We assumed that the reduction of the test-retest time period from 12 weeks to 1 week in our study resulted in increased test-retest reliability.

The follow-up and coordination scale had the lowest mean score, while delivery system design/decision support had the highest mean score. These results demonstrate clearly that while the healthcare system for ESRD patients in Korea may be physician-centered. Although the decision support of the CCM is intended to be physician-oriented, it was said that the questions about decision support correspond to the evaluation of self-management in CCM [13]. The follow-up/coordination subscale is important for CCM elements that include self-management support [6]. Almost all PACIC subscales are measures of self-management support.

While more than 1000 healthcare organizations have participated in health care improvement in the USA, the CCM is not yet a well-established concept in Korea. However, the overall scores, as well as those of the five scales in this study, were similar to those in the validation study of Glasgow et al. [6]. Most of the scales were rated higher in our study than in another study in osteoarthritis patients [13]. This may be because the care for patients with ESRD puts more emphasis on key elements of self-management care and planned, proactive care compared to care for other chronic diseases.

These results showed that increased patient satisfaction with staff was associated with improved self-efficacy. Self-efficacy is confidence in managing chronic disease, so, PACIC scores can significantly and substantially predict provider support for self-management.

The relationship between patient and clinicians correlated with serum potassium levels in hemodialysis patients in previous studies. Serum potassium was significantly associated with dietary self-efficacy [10]. Surprisingly, we found that increased patient activation and problem solving and total PACIC scores were associated with lowered serum potassium. These outcomes had complex interactions with patient–clinician bonds and with patient perceptions of the dialysis center. Improved patient perceptions of the dialysis center and characteristics of the dialysis center such as solidarity were also associated with improved patient compliance and were assessed by a composite laboratory measure. Hemodialysis patients in highly managed relationships might have improved self-efficacy, especially improved dietary self-efficacy, which have good effect on the serum potassium. Patients with increased dietary self-efficacy have lower serum potassium, show better adherence to attitudes and behaviors toward prescribed regimens and have better relationships with medical staff [12].

PACIC was also significantly associated with patients’ HRQOL. In other words, patient-centered care has good results on the health outcomes of patients. Patient’s satisfaction with care had a good effect on health outcomes [14,15], and effective communication would have helped the patient’s decision-making about treatment as a result. Several studies [16,17] suggest that decision coaching in conjunction with decision aids may be effective in increasing decision-making participation, increasing knowledge, aligning decisions with patient values, and reducing decision-making conflicts. A systematic review of interventions to support decision-making suggests that decision coaching (i.e., individualized and facilitated discussions to prepare patients for upcoming decisions) provides information about the decision-making process, coordination of decisions between caregivers and patients, and options [16,17]. Communication in patients with CKD can be an effective way to engage patients and allow them to safely express their personal experiences and feelings.

The PACIC evaluates the components of CCM: the interactive and comprehensive support for chronic disease treatment provided by health care providers. In particular, the Institute of Medicine suggested the need for a ‘patient-centered’ approach to the management of chronic disease patients, and for this reason, an appropriate questionnaire was needed to evaluate practical support for chronic disease from the patient’s point of view.

The CCM is not yet a well-established concept in Korea. However, the overall scores, as well as those of the five scales in this study, were similar to those in the validation study of Glasgow et al. [6]. Most of the scales were rated higher in our study than in another study in osteoarthritis patients [18]. This may be because the care for patients with ESRD puts more emphasis on key elements of self-management care and planned, proactive care compared to care for other chronic diseases.

Although patients with other chronic medical conditions may view their medical care differently than those with diabetes, the PACIC, which was tested in populations with various chronic conditions, shows no differences in its psychometric properties for different conditions. This study was performed in patients with ESRD, so the results cannot be generalized to patients with other chronic conditions. There was a limitation in that the study subjects were selected only from two dialysis centers and could not represent all hemodialysis patients. However, one strength of this study is that the Korean version of PACIC was developed through a systematic translation and application process, and reliability and validity were verified. In particular, it was found that PACIC was also associated with the health outcomes of ESRD patients.

Our results confirmed that the Korean-translated PACIC questionnaire can be used in hemodialysis patients. Despite the differences in culture and healthcare systems, the Korean-translated PACIC retained excellent psychometric properties. Additional validation studies in other independent samples will allow this tool to be used to assess patient–clinician relationships (treatment satisfaction) for chronic diseases, including ESRD, in various clinical settings. In summary, comprehensive treatment strategies are needed to improve health outcomes as well as self-efficacy for disease management in ESRD patients.

## Figures and Tables

**Table 1 healthcare-10-01149-t001:** Patient characteristics and overall PACIC scores.

Characteristics	Number (%)	PACIC (Mean ± SD)	*p* Value *
Sex			
Male	91 (52.9)	3.1 ± 0.8	0.098
Female	81 (47.1)	2.9 ± 0.8	
Age (mean ± SD (years))	59.2 ± 12.9		
≤49 years	40 (23.3)	2.8 ± 0.8	0.329
50–59 years	44 (25.6)	2.9 ± 0.8	
60–69 years	46 (26.7)	3.1 ± 0.8	
≥70 years	42 (24.4)	3.1 ± 0.8	
Educational level			
Elementary or less	35 (20.3)	2.8 ± 0.7	0.667
Middle school	33 (19.2)	3.0 ± 0.9	
High school	59 (34.3)	3.0 ± 0.7	
College or more	45 (26.2)	3.0 ± 0.8	
Job status			
No	123 (71.5)	2.9 ± 0.8	0.278
Yes	49 (28.5)	3.1 ± 0.8	
Dialysis duration			
<2 years	65 (37.8)	3.0 ± 0.8	0.866
2–5 years	68 (39.5)	2.9 ± 0.9	
>5 years	39 (22.7)	3.0 ± 0.9	
Cause of ESRD			
Hypertension	51 (29.7)	2.8 ± 0.9	0.432
Diabetes	68 (39.5)	3.0 ± 0.8	
Glomerulonephritis	18 (10.5)	3.1 ± 0.7	
Other	35 (20.3)	3.0 ± 0.7	
Co-morbidity (yes)			
Hypertension	145 (84.3)	3.0 ± 0.9	0.969
Diabetes	89 (51.7)	2.9 ± 0.8	0.458
Stroke	20 (11.6)	3.0 ± 0.9	0.725
Coronary disease	17 (9.9)	2.9 ± 0.9	0.672
Serum potassium (mEq/L)			
<5.0	89 (51.7)	3.1 ± 0.7	0.042
≥5.0	83 (48.3)	2.8 ± 0.8	
Serum phosphorus (mEq/L)			
<4.5	57 (33.1)	2.9 ± 0.8	0.341
≥4.5	115 (66.9)	3.0 ± 0.8	
Serum albumin (g/dL)			
<4.0	99 (57.6)	3.0 ± 0.8	0.837
≥4.0	73 (42.4)	3.0 ± 0.8	
Interdialytic weight gain (kg)			
<2.0	34 (19.8)	2.9 ± 0.7	0.776
≥2.0	138 (80.2)	3.1 ± 0.8	
Total	172(100.0)	3.0 ± 0.8	

* *p* values were determined by *t*-test and ANOVA.

**Table 2 healthcare-10-01149-t002:** Descriptive statistics, score distributions and internal reliability of a Korean translation of the PACIC survey scale in patients on hemodialysis.

Characteristics	Mean ± SD	Ceiling Effects	Floor Effects	Cronbach’s Alpha	Item-Scale Correlation	Test-Retest Correlation (ICC)
		(%)	(%)			
PACIC	3.0 ± 0.8	0.6	0.0	0.83		0.82
Patient activation	3.4 ± 0.8	3.5	0.6	0.65	0.67–0.79	0.75
Delivery system design/decision support	3.4 ± 0.9	6.4	1.7	0.70	0.77–0.81	0.79
Goal setting/tailoring	2.8 ± 0.9	2.9	3.5	0.81	0.72–0.79	0.72
Problem solving/contextual	3.1 ± 1.0	5.2	4.7	0.84	0.79–0.85	0.74
Follow-up/coordination	2.5 ± 1.0	1.7	5.8	0.82	0.72–0.78	0.81

**Table 3 healthcare-10-01149-t003:** Association of dietary self-efficacy, serum potassium, HRQOL and PACIC by the multiple (logistic) regression analysis.

Characteristics	Dietary Self-Efficacy				Potassium				HRQOL			
	b	SE	Standardized B	*p* Value	**b**	SE	OR	*p*-Value	b	SE	Standardized B	*p* Value
Overall PACIC score	0.22	0.06	0.27	0.000	0.50	0.24	1.65	0.040	7.29	0.95	0.50	<0.001
Patient activation	0.20	0.06	0.27	0.000	0.49	0.23	1.63	0.037	5.65	0.95	0.41	<0.001
Delivery system/practice design	0.17	0.05	0.23	0.002	0.20	0.21	1.22	0.340	4.58	0.93	0.35	<0.001
Goal setting/tailoring	0.12	0.05	0.17	0.027	0.29	0.21	1.34	0.155	4.70	0.88	0.38	<0.001
Problem solving/contextual	0.20	0.05	0.31	0.000	0.43	0.19	1.54	0.022	5.29	0.78	0.46	<0.001
Follow up/coordination	0.13	0.05	0.19	0.011	0.38	0.20	1.46	0.062	5.69	0.77	0.48	<0.001

Individually adjusted for sex, age, educational level, job, dialysis duration, cause of ESRD, and comorbidity; HRQOL: Health-related quality of life, OR: odds ratio.

## Data Availability

Data available on request due to restrictions (privacy). The data presented in this study are available on request from the corresponding author. The data are not publicly available due to privacy.

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
