# Peer review of "Psychometric Properties and Effects on Health Outcomes of the Patient Assessment of Chronic Illness Care (PACIC) in Korean Hemodialysis Patients"

_healthcare, 2022, doi:10.3390/healthcare10061149_

Round 1

Reviewer 1 Report

Dear authors,

Thanks for the opportunity to review the manuscript. A very interesting manuscript considering the lack of publications in this area. It is recommended to add the proposed validated version of the scale as an appendix.

Author Response

yes. Added questionnaire to appendix

Reviewer 2 Report

The title of the study does not correspond to the study.

The objective stated in the text does not either, but the objective stated in the abstract does not, so it should be modified in the text and put the one in the abstract, only one objective not objectives: 

Our aim were to evaluate the psychometric properties of the Korean-translated patient assessment of chronic illness care (PACIC) in patients with ESRD, to evaluate the associations between patient assessment about relationship with staff and health outcomes among patients

It is not explained how the first sample has been reocgured, if in the second but it is not known where the patients come from and the reason for this number.

This conclusion does not correspond to the design of the article:

In summary, these results indicate that for patients with 226 ESRD, patient-staff relationships (treatment satisfaction) may be an important modifiable 227 mediator to improve outcomes such as confidence in dealing with dietary problems and 228 clinical outcome.

Author Response

Attached file

Reviewer 3 Report

Thank you for the opportunity to review the manuscript 1745555 “The Effects on Health Outcomes of the Patient-Centeredness of Care in Korean Hemodialysis Patients.”

The aims of the paper were to evaluate the psychometric properties of the Korean-translated patient assessment of chronic illness care (PACIC) in patients with ESRD and to  evaluate the associations between patient assessment about relationship with staff and health outcomes among patients.

The introduction takes up the aspect of questionnaires. The chronic care model is also addressed. A theoretical background supported by references is missing. This should focus on the aims of the work and provide the basis for discussion of the results. Please combine then the introduction with your results.

The discussion needs to be revised. There is no connection to the background. There is no in-depth analysis of the literature. Limitations of the work are missing.

Author Response

Attached file

Round 2

Reviewer 2 Report

The corrections would be sufficient for publication

This manuscript is a resubmission of an earlier submission. The following is a list of the peer review reports and author responses from that submission.